# Evaluation of the Antimicrobial Activity of a Formulation Containing Ascorbic Acid and Eudragit FS 30D Microparticles for the Controlled Release of a Curcumin–Boric Acid Solid Dispersion in Turkey Poults Infected with *Salmonella enteritidis*: A Therapeutic Model

**DOI:** 10.3390/ijms242216186

**Published:** 2023-11-10

**Authors:** Daniel Hernandez-Patlan, Bruno Solis-Cruz, Juan D. Latorre, Jesus A. Maguey-Gonzalez, Inkar Castellanos-Huerta, Eric Beyssac, Ghislain Garrait, Alma Vázquez-Durán, Raquel López-Arellano, Abraham Méndez-Albores, Billy M. Hargis, Guillermo Tellez-Isaias

**Affiliations:** 1Laboratory 5: LEDEFAR, Multidisciplinary Research Unit, Superior Studies Faculty at Cuautitlan (FESC), National Autonomous University of Mexico (UNAM), Cuautitlan Izcalli 54714, Mexico; lopezar@unam.mx; 2Nanotechnology Engineering Division, Polytechnic University of the Valley of Mexico, Tultitlan 54910, Mexico; 3Division of Agriculture, Department of Poultry Science, University of Arkansas, Fayetteville, AR 72701, USA; jl115@uark.edu (J.D.L.); jm201@uark.edu (J.A.M.-G.); icastell@uark.edu (I.C.-H.); bhargis@uark.edu (B.M.H.); gtellez@uark.edu (G.T.-I.); 4UFR Pharmacie, UMR MEDIS, Université Clermont-Auvergne, F-63001 Clermont-Ferrand, France; eric.beyssac@uca.fr (E.B.); ghislain.garrait@uca.fr (G.G.); 5Unidad de Investigación Multidisciplinaria L14 (Alimentos, Micotoxinas y Micotoxicosis), Facultad de Estudios Superiores Cuautitlán, Universidad Nacional Autónoma de México, Cuautitlán Izcalli 54714, Mexico; almavazquez@comunidad.unam.mx (A.V.-D.); albores@unam.mx (A.M.-A.)

**Keywords:** *S. enteritidis*, Eudragit FS 30D, curcumin, boric acid, ascorbic acid, interaction studies, release studies, degradation studies

## Abstract

The selection of components within a formulation or for treatment must stop being arbitrary and must be focused on scientific evidence that supports the inclusion of each one. Therefore, the objective of the present study was to obtain a formulation based on ascorbic acid (AA) and Eudragit FS 30D microparticles containing curcumin–boric acid (CUR–BA) considering interaction studies between the active components carried out via Fourier transform infrared spectrometry (FTIR) and differential scanning calorimetry (DSC) to minimize antagonistic effects, and comprehensively and effectively treat turkey poults infected with *Salmonella enteritidis* (*S. enteritidis*). The DSC and FTIR studies clearly demonstrated the interactions between AA, BA, and CUR. Consequently, the combination of AA with CUR and/or BA should be avoided, but not CUR and BA. Furthermore, the Eudragit FS 30D microparticles containing CUR–BA (SD CUR–BA MP) showed a limited release of CUR–BA in an acidic medium, but they were released at a pH 6.8–7.0, which reduced the interactions between CUR–BA and AA. Finally, in the *S. enteritidis* infection model, turkey poults treated with the combination of AA and SD CUR–BA MP presented lower counts of *S. enteritidis* in cecal tonsils after 10 days of treatment. These results pointed out that the use of an adequate combination of AA and CUR–BA as an integral treatment of *S. enteritidis* infections could be a viable option to replace the indiscriminate use of antibiotics.

## 1. Introduction

*Salmonella enteritidis* (*S. enteritidis*) is one of the most important members of the various Salmonella serovars (about 2500) that can be transmitted from poultry and its by-products to humans, causing serious health problems [1]. Therefore, its prevention and/or control is of utmost importance, not only at a health level, but also at a production level since it has important implications such as alterations in production parameters and even death in the poultry, which results in significant economic losses [2,3]. To mitigate these problems, the poultry industry has indiscriminately included antibiotics into diets in order to promote growth and prevent bacterial diseases for a long time [4]. However, due to this overuse of antibiotics for prophylactic and growth promotion purposes in poultry production, their current status in some European, American, and Asian countries is prohibited [5]. In this context, many efforts are being made to prevent and control *Salmonella* infections through promising alternatives to antibiotics such as the use of probiotics, prebiotics, organic acids, essential oils, and phytobiotics, among others [1,6,7]. However, those that have presented the best results are probiotics and organic acids, followed by phytobiotics in recent years [8].

Although it has been described that supplementation with alternatives to antibiotics can positively affect performance and health in poultry due to their antimicrobial, antioxidant, immune system stimulant, or intestinal health promotion properties [8], the vast majority of these compounds produce inconsistent results and rarely match antibiotics in their effectiveness [9]. Consequently, the use of combinations of alternatives to antibiotics could be the key to maximize performance and poultry production, as well as improving or potentiating antimicrobial effects [10].

Considering this background, in the present study, three alternatives to the use of antibiotics were selected: ascorbic acid (AA), a gluconic acid lactone derived from glucuronic acid, and ketolactone (organic acid), which has antioxidant and antimicrobial properties, as well as immodulatory functions [11]. Boric acid (BA) is a weak boron acid of inorganic nature that has shown good antimicrobial activity against *S. enteritidis* in previous studies conducted in our laboratories [12,13]. Curcumin (CUR) is a phytobiotic with excellent antioxidant, anti-inflammatory, and antimicrobial properties, but poor bioavailability derived from its low solubility [14], which has been improved through its formulation in solid dispersions [13]. The selection of these alternatives was based on previous studies published by our laboratories in which it was shown that AA presented excellent antimicrobial properties by reducing *S. enteritidis* counts, but in the presence of BA and CUR, its antimicrobial effect was reduced because antagonistic interactions occurred [12]. In contrast, the interaction between CUR formulated in a solid dispersion with PVP K30 and BA showed a significant increase in its antimicrobial activity against *S. enteritidis*, that is, synergistic effects were observed [13].

The present study is a continuation of a series of articles published by our research group and aimed to obtain a formulation based on ascorbic acid (AA) and Eudragit FS 30D microparticles containing curcumin-boric acid (CUR-BA) considering interaction studies between the active components carried out via Fourier transform infrared spectrometry (FTIR) and differential scanning calorimetry (DSC) to minimize antagonistic effects, and comprehensively and effectively treat turkey poults infected with *Salmonella enteritidis* (*S. enteritidis*).

## 2. Results

### 2.1. Compatibility Studies

#### 2.1.1. ATR FT-MIR Spectroscopy

The IR spectra of the pristine components, as well as the four mixtures, are shown in Figure 1. In the IR spectra of pure ascorbic acid (AA, Figure 1A), stretching vibrations from 3524 cm^−1^ to 3000 cm^−1^ correspond to its different hydroxyl groups (O–H). In the case of the band at 2914 cm^−1^, this corresponds to the C–H stretch. In addition, the absorption bands found at 1753 cm^−1^, 1655 cm^−1^, and 1314 cm^−1^ are related to C=C stretching vibrations of the five-membered lactone ring. Likewise, the absorption bands corresponding to the stretching vibrations of C–O–C, C–C, and C–O–H are found at 1113 cm^−1^, 1023 cm^−1^, and 987 cm^−1^, respectively.

In the case of the IR spectrum of curcumin (CUR, Figure 1A), the absorption bands corresponding to the stretching of the O–H bonds are found at 3506 cm^−1^ and 3313 cm^−1^. On the other hand, C=O and C=C, as well as the vibrational stretch of C=C, are located at the bands of 1626 cm^−1^ and 1507 cm^−1^, respectively. Another important band within the IR spectrum of curcumin is at 1601 cm^−1^ and corresponds to the vibrational stretching of the C=C_ring_. In addition, the characteristic bands of the C–O and C–O–C bonds are located at 1273 cm^−1^ and 1113 cm^−1^.

When analyzing the IR spectrum of the AA–CUR (1:1 weight ratio) mixture (Figure 1A), it is possible to appreciate that the bands related to the hydroxyl groups of ascorbic acid (3524–3207 cm^−1^), as well as the band corresponding to the stretching vibration C=O of the five-membered lactone ring (1753 cm^−1^) are reduced in intensity. Furthermore, a notable reduction in the intensity of the characteristic bands to the C=C ring of AA (1655 cm^−1^ and 1314 cm^−1^) is observed.

Figure 1B shows the IR spectrum of boric acid (BA), in which the characteristic band of the hydroxyl groups is at 3188 cm^−1^. Furthermore, the band at 1401 cm^−1^ corresponds to the asymmetric stretching of B–O in the chemical structure of BA (BO_3_)^3−^. The band found at 1192 cm^−1^ is due to atomic vibrations in the B–OH plane bending. The weak band at 882 cm^−1^ represents the B–O_symmetric_ stretching vibration. 

Analyzing the IR spectrum of the BA–CUR (1:1 weight ratio) mixture (Figure 1B), some of the characteristic bands of CUR disappeared, such as those corresponding to the hydroxyl groups (3313 cm^−1^) and the C=O carbonyl groups (1507 cm^−1^). Furthermore, the characteristic BA bands found at 1401 cm^−1^ (B–O_asymmetric_) and 1192 cm^−1^ (B–OH) were reduced in intensity in the mixture, presenting a behavior more similar to that of CUR.

Unlike the previous mixtures, the AA–BA (1:1 weight ratio) mixture (Figure 1C) shows more marked differences since the vibrational stretching of O–H (3524 cm^−1^ to 3000 cm^−1^), C=O (1753 cm^−1^), C=C_ring_ (1655 cm^−1^ and 1314 cm^−1^), C–O–C (1113 cm^−1^), C–C (1023 cm^−1^), and C–O–H (987 cm^−1^) of AA were reduced in intensity, as well as in the case of the bands corresponding to the B–O_Asymmetric_ (1401 cm^−1^) and B–OH (1192 cm^−1^) of BA (Figure 1C).

Finally, Figure 1D shows the IR spectrum of the AA–BA–CUR (1:1:1 weight ratio) mixture, which is clearly modified with respect to the individual IR spectra of AA, BA, and CUR, and this is related to those observed in the binary mixtures; that is, the vibrational stretches of C=C_ring_ (1655 cm^−1^ and 1314 cm^−1^) of AA disappear, like those corresponding to B–O_Asymmetric_ (1401 cm^−1^) and B–OH (1192 cm^−1^) from BA. In the case of CUR, the intensities in its main bands are diminished. 

#### 2.1.2. Differential Scanning Calorimetry (DSC)

Differential scanning calorimetry (DSC) was used as a tool to compare the thermal behavior of AA, BA, and CUR, as well as AA–CUR (1:1 weight ratio), BA–CUR (1:1 weight ratio), AA–BA (1:1 weight ratio), and AA–BA–CUR (1:1:1 weight ratio) mixtures (Figure 2).

The DSC curve of AA shows the melting peak at 193.38 °C with an enthalpy of 238.87 J/g, and after 200 °C, AA begins to decompose (Figure 2A). Otherwise, CUR exhibits its melting peak at 173.47 °C with an enthalpy of 36.63 J/g (Figure 2A). However, the thermal behavior of the combination of AA and CUR (1:1 weight ratio) was modified, since the melting peak of AA reached 190.12 °C, the onset temperature changed from 192.60 °C to 183.91 °C, the enthalpy reduced to 191.57 J/g and from the melting point temperature, AA begins to degrade (Figure 2A). In the case of CUR, the melting temperature reached 171.91 °C and its onset temperature changed from 161.14 °C to 163.25 °C; however, the enthalpy remained very close to that of raw CUR (39.19 J/g). In Figure 2B, it can be seen that the thermal behavior of the BA–CUR (1:1 weight ratio) mixture is completely different compared with the independent DSC curves of BA and CUR. In this context, the endotherm of CUR disappeared and those corresponding to BA were modified, since their melting points shifted from 153.72 °C to 148.38 °C and from 174.25 °C to 162.66 °C, respectively, as well as the onset temperatures and enthalpies (Figure 2B), which suggests a strong interaction between both components. 

In the AA–BA mixture (1:1 weight ratio), changes in the fusion isotherms of the components were clearly observed, which suggests a strong interaction between them, and even above 130 °C, the decomposition processes begin to be noticeable (Figure 2C). Specifically, the endotherms found at 153.72 °C and 174.25 °C of BA corresponding to dehydration processes were displaced to 138.13 °C with a change in its onset temperature from 144.56 °C to 135.13 °C and modification in the enthalpy from 859.63 J/g to 116.22 J/g, and 144.29 °C with a change in the onset temperature from 170.51 °C to 144.17 and modification in the enthalpy of 200.28 J/g to 71.33 J/g, respectively. Furthermore, the AA melting peak disappeared in the DSC curve of the AA–BA mixture.

In the case of the mixture that contains the three components, AA–BA–CUR (1:1:1 weight ratio), it can be observed again that the melting peak corresponded to AA (193.38 °C) and CUR (173.47 °C), since the endotherm that is found at 169.19 °C, with an onset temperature of 162.82 °C and enthalpy of 74.80 J/g, suggests that it is the second dehydration of BA (Figure 2D). Furthermore, the thermal behavior corresponding to the first dehydration of BA was modified, since its temperature moved from 153.72 °C to 144.47 °C, as well as the onset temperature from 144.56 °C to 128.98 °C. Likewise, the enthalpy was reduced from 859.63 J/g to 532.07 J/g (Figure 2D). These changes in thermal behavior suggest that CUR has a protective effect as it gives stability to BA, since the thermogram of the AA–BA mixture (1:1 weight ratio) shows that the melting peak of AA disappears, and BA endotherms move and begin to degrade (Figure 2C).

### 2.2. Characterization of Eudragit FS 30D Microparticles and Release Studies

Once the process of obtaining the SD CUR–BA was standardized, as well as the selection of the components and proportions of the Eudragit FS 30D polymeric coating and the optimization of their production process, the resulting microparticles (SD CUR–BA MP) were characterized in terms of particle size and release of CUR–BA. In general, Eudragit FS 30D microparticles of SD CUR–BA had a mean particle size of 680 µm. 

The release profile of SD CUR–BA in pH 5.2, 1.2, 6.8, and 7.0 media from Eudragit FS 30D microparticles (SD CUR–BA MP) is presented in Figure 3. The release of CUR–BA in the first release medium (pH 5.2) was 3.2% at 30 min, followed by 5.4% in the pH 1.2 medium at 45 min, 8.8% in the pH 6.8 medium at 45 min and finally 11.2% in the last release medium (pH 7.0) at 75 min; that is, the total release was 28.6% at 195 min.

### 2.3. Degradation Studies of SD CUR–BA at Different pH

The results of the degradation profile of SD CUR–BA performed in the USP IV apparatus in buffer solutions of hydrochloric acid pH 1.2, acetates pH 5.2, and phosphates pH 6.8 and 7.0 at 42 °C are shown in Figure 4. In Figure 4A, it can be seen that the concentration of SD CUR–BA was reduced by 45.6% at pH 6.8–7.0 at 120 min, while at pH 1.2 and 5.2 it was reduced by 21.5% and 13.2, respectively, when the concentration of PVP K30 was 0.3 mg/mL. Analyzing the results of Figure 4B at 120 min, the reduction in the concentration of SD CUR–BA when the concentration of PVP K30 was 0.6 mg/mL was 32.8%, that is, 12.8% less compared with the results obtained when the PVP K30 concentration was 0.3 mg/mL, but at pH 1.2 and 5.2 the degradation was similar to that when the PVP K30 concentration was 0.3 mg/mL. Finally, when the concentration of PVP K30 was 1.2 mg/mL, the concentration of SD CUR–BA was 17.0%, 17.6% and 21.5% at pH 5.2, 6.8–7.0 and 1.2, respectively at 120 min (Figure 4C). These results suggest that the degradation of SD CUR–BA at pH 1.2 is the same regardless of the concentration of PVP K30, but at pH 6.8–7.0, the degradation decreases as the concentration of PVP K30 increases.

### 2.4. In Vivo Studies

The results of the therapeutic administration of AA (0.033%), SD CUR–BA MP (0.067%), and AA/SD CUR–BA MP (0.1%) into the feed on the counts of *S. enteritidis* in the crop and cecal tonsils (CT) after 3 and 10 d of treatment are summarized in Table 1. Although no significant reduction in *S. enteritidis* counts in the crop was found after 3 d of treatment in infected turkey poults compared with CTR (+), at 10 d of treatment, the turkey poults treated with AA showed a significant reduction in *S. enteritidis* counts (0.73 log_10_ cfu/g, *p* < 0.05) compared with CTRL (+). Instead, *S. enteritidis* counts in CT after 3 d of treatments were significantly decreased in turkey poults treated with AA/SD CUR–BA MP (0.60 log_10_ cfu/g, *p* < 0.05) compared with CTRL (+). Regarding the counts of *S. enteritidis* at 10 d of treatment, turkey poults treated with SD CUR–BA MP and AA/SD CUR–BA MP had significantly lower counts compared with CTRL (+) (0.87 and 1.03 log_10_ cfu/g, respectively, *p* < 0.05). However, in poults treated with AA/SD CUR–BA MP, *S. enteritidis* counts were reduced by 0.16 log_10_ cfu/g more compared with turkey poults treated with SD CUR–BA MP.

Table 2 shows the weights of the poults at the beginning of the experiment (BW D_0_) until day 10 (BW D_10_), and the body weight gained from D_0_ to D_10_ (BWG). Even though the turkey poults that were treated with AA and AA/SD CUR–BA MP had a significantly lower initial weights (*p* < 0.05) compared with the turkey poults in the other groups, only the weight of the group treated with AA/SD CUR–BA BA MP on day 10 was significantly higher compared with CTRL (+) (*p* < 0.05), followed by the group of turkey poults treated with SD CUR–BA MP (*p* < 0.05). Furthermore, turkey poults treated with SD CUR–BA MP or AA/SD CUR–BA MP showed a significantly higher BWG compared to CTRL (+) (*p* < 0.05). On the other hand, turkey poults treated with AA had a higher weight compared with CTRL (+) and CTRL (−), but not significantly.

Finally, the serum concentrations of FITC-d in turkey poults treated with AA, SD CUR–BA MP, and AA/SD CUR–BA MP were significantly lower compared with turkey poults from the CTRL (+) group. However, the serum concentrations of FITC-d in turkey poults treated with SD CUR–BA MP and AA/SD CUR–BA MP did not show significant differences with respect to CTRL (−).

## 3. Discussion

The current trend in the use of antibiotics in poultry production due to bacterial resistance problems has led to the investigation and use of alternative additives to replace antibiotics [4]. These feed additives are non-nutritive natural products that are added as minor components of the diets of animals, since it has been described that they can improve growth performance and feed conversion rate, nutrient digestibility, maintain homeostasis in the intestinal microbiome, and stimulate the immune system [8,15]. Among the most used feed additives in poultry farming are phytobiotics such as essential oils, oleoresins, and herbal extracts, among others [16], as well as probiotics and prebiotics [17], organic and inorganic acids [18], enzymes [19], and phages and antimicrobial peptides [4].

Despite the fact that a large number of feed additives have been studied and evaluated as a replacement for the use of antibiotics, most of them are relatively effective and economically feasible, as well as variable in terms of the effects they produce [20]. In this context, one of the strategies that have been considered in order to solve the problems of effectiveness of feed additives is the use of combinations of two or more of them in the same treatment [21,22,23,24]. However, the selection of the additives for their combination should stop being arbitrary or based on empirical knowledge; that is, the tendency is always that the combinations of feed additives present synergistic interactions to enhance their effects, which can also favor the reduction in treatment costs derived from the lowest doses.

Therefore, in the first part of this research, studies of interactions between different binary mixtures were carried out considering feed additives such as AA, BA, and/or CUR from characterization techniques such as FTIR spectrometry and DSC. According to the results, the changes in FTIR spectra prove the interaction between AA and CUR (1:1 weight ratio), since the bands related to the C=C stretching vibrations of the five-membered lactone ring (C=C_ring_), which are located at 1655 cm^−^^1^ and 1314 cm^−^^1^, respectively, were modified (Table 1). Likewise, the thermal behavior (DSC curve) of the said mixture showed a decrease in melting temperatures, as well as modifications in onset temperatures with respect to the pure components. Regarding the enthalpies, only that corresponding to AA decreased from 238.87 J/g to 191.57 J/g, and AA also began to degrade around 192.12 °C (Figure 2A). Considering these results, a strong interaction between the components is suggested, which indicates antagonistic interactions between AA and CUR, since a study previously performed in our laboratories, in which the antimicrobial activity of 1% of the AA–CUR (1:1 weight ratio) mixture was evaluated against *S. enteritidis* in an in vitro model simulating the crop, proventriculus, and intestine of broiler chickens showed no effect in reducing *S. enteritidis* counts in any of the simulated sections, but AA by itself was able to reduce *S. enteritidis* counts in the crop-simulating section [12]. Furthermore, these results are supported by a study in which it was estimated that the mixture of AA and CUR should be avoided in formulations due to their strong interaction [25], as well as in other study where antagonistic interactions were established in the antioxidant capacity of CUR in the presence of AA [26]. The mechanism by which these antagonistic interactions between AA and CUR occur is given by the oxidation of AA to dehydroascorbic acid by the action of the phenoxyl radical of CUR, which leads to the loss of antioxidant activity [27].

Like AA–CUR (1:1 weight ratio), the IR spectrum of the BA–CUR (1:1 weight ratio) mixture was modified with respect to the individual IR spectra of BA and CUR (Figure 1B). In fact, the most important changes occurred in the bands of the vibrational stretching of the hydroxyl groups of curcumin at 3313 cm^−^^1^ and the C=O carbonyl groups (1507 cm^−^^1^), as well as in the B–O_Asymmetric_ bands of BA at 1401 cm^−^^1^, which suggests the formation of rosocyanine, a complex of which the center is a boron tetrahedral atom and two curcumin molecules, and/or rubrocurcumin, a red complex formed by one curcumin molecule, one boron molecule, and an oxalate ligand [28,29]. Furthermore, the thermal behavior of the mixture was also considerably affected (Figure 2B). It has been described that these complexes formed between boron and CUR have excellent antimicrobial, anticancer and antifungal properties, since the stability of CUR and its biological activity are improved [30]; that is, this interaction was synergistic. These synergistic interactions were also previously described [12], concluding that BA and the combination between BA and CUR (1:1 weight ratio) had an excellent effect, since they completely eliminated *S. enteritidis* in the simulated intestinal section in an in vitro avian model, but they had no effect on the sections corresponding to the crop and proventriculus [12].

Regarding the AA–BA mixture (1:1 weight ratio), there was evidently a change in its IR spectrum compared with the individual components, since the intensity in the most important AA and BA bands decreased (Figure 1C), which suggests a strong interaction between both components, leading to the hydrolytic instability of the complexes formed between AA and BA [31]. Similarly, the thermal behavior of the AA–BA mixture (1:1 weight ratio) (Figure 2C) was consistent with previous results due to the corresponding DSC curve, which suggests that the decomposition processes began at 130 °C, which supports the results of the antimicrobial activity of the AA–BA (1:1 weight ratio) mixture against *S. enteritidis* previously obtained in an in vitro model, since its activity was null in the simulated sections (crop, proventriculus, and intestine) [12].

Based on the interactions of the binary mixtures, the mixture of the three components (AA–BA–CUR) is not a viable option as a possible treatment, since the IR spectrum (Figure 1D) and the DSC curve (Figure 2D) indicate strong interactions, which compromise antimicrobial activity, as previously seen in the in vitro digestibility model [12]. Therefore, considering the antimicrobial potential of the BA–CUR (1:1 weight ratio) mixture and AA independently, as a second part of the present study, a solid dispersion of CUR–BA (SD CUR–BA, 1:1) was prepared, since it previously showed promising results by reducing the *S. enteritidis* counts in an in vitro model that simulates the gastrointestinal conditions of broiler chickens and in a broiler prophylactic model [13]. In the case of AA, in a prophylactic model, *S. enteritidis* counts were significantly reduced in the crop [32], whereas in a therapeutic model of *S. enteritidis* infection in broiler chickens, bacterial counts were decreased in both crop and cecal tonsils [32].

In this context, with the aim of having an integral treatment considering AA and SD CUR–BA to have an antimicrobial effect in both the crop and intestine (specifically in cecal tonsils), controlled release microparticles of SD CUR–BA were prepared using Eudragit FS 30D (SD CUR-BA MP) to avoid interactions between the components, since it is a pH-dependent anionic copolymer of methacrylic acid, methyl acrylate and methyl methacrylate for drug release in the colon (pH ≥ 7) [33], but it has been reported that the release of drugs can begin from pH 6.8 [34,35]. Then, taking into account that the crop is the first section of the gastrointestinal tract and that, in addition, its functions is the ability to store feed and the beginning of fermentation processes by lactobacilli, it has been reported that its pH may be slightly below 5 or above 6 in broiler chickens and turkeys [36,37,38,39], which would ensure the avoidance of interactions between the components of the formulation.

Analyzing the release profile of SD CUR–BA MP (680 µm average particle size) (Figure 3), it was possible to establish that the released percentage of SD CUR–BA was 3.2% in 30 min at pH 5.2, average pH and time established for crop conditions [36,40], which is a low percentage that was achieved through the 17% increase in solids and the use of 15% plasticizer, as well as considering that Eudragit FS 30D is soluble from pH 7, but insoluble at acidic pH [41]. These results are consistent with other studies where the release of bacteria from Eudragit FS 30D microparticulate systems was less than 5% at an acidic pH [42] and less than 10% in microparticles containing glutathione and S-nitrosoglutathione at pH less than 6 [33]. Furthermore, if the amount of SD CUR–BA released after 45 min at pH 1.2 is considered, the total cumulative release was 8.6% (75 min), which would be in agreement with one of the studies mentioned above [33]. However, the greatest release of SD CUR–BA occurred from the pH change of 6.8 and 7.0, since the release was 20% after 120 min. It is important to mention that 100% release of SD CUR–BA was not reached since CUR is a very unstable molecule at a pH above 6; in fact, it is known that at a pH 6.8, it degrades around 30% and at a pH 7.0 approximately 50%, both after 60 min, but at an acidic pH, it is very stable, degrading only 1% at pH a 1.2 after 6 h [43]. 

Although the degradation of CUR increases as the pH increases and its biological properties are compromised, it is a fact that PVP K30 gives it stability while increasing the dose, as shown in Figure 4. These results are consistent with studies that have shown that the degradation of CUR and indomethacin are improved as the proportion of PVP in solid dispersions increases [44,45]. Likewise, it is known that in the case of CUR, the formation of complexes with boric acid gives it greater stability [30]. 

Considering the antimicrobial effect of ascorbic acid in reducing the counts of *S. enteritidis* in the crop in a previous study in broiler chickens [32] and having understood the release behavior of SD CUR–BA from the Eudragit FS 30D microparticles, which guarantees that interactions between the components that could lead to antagonistic effects are minimized, the antimicrobial effect of AA (0.033%), SD CUR-BA MP (0.067%), and the combination of AA and SD CUR–BA MP (0.1%) was evaluated in a therapeutic administration model in turkey poults for 3 and 10 days post *S. enteritidis* challenge (Table 1). The results clearly show that AA has a significant antimicrobial effect at the crop level compared with CTRL (+) at 3 and 10 d of treatment (*p* < 0.05), and although there were no significant differences in the counts of *S. enteritidis* in the crop of turkey poults supplemented with AA/SD CUR–BA MP on day 10 of treatment compared with CTRL (+), it was clearly observed that *S. enteritidis* counts were reduced by 0.67 log_10_ cfu/g. The antimicrobial effect of AA is mainly due to its ability to reduce the pH in the medium, providing inadequate conditions for bacteria to survive [32,46]. Furthermore, it is clear that there were no interactions between the components of the formulation that compromised the antimicrobial activity of AA. 

Analyzing the results in cecal tonsils (Table 1), it can be observed that in turkey poults administered AA/SD CUR–BA MP, *S. enteritidis* counts were significantly reduced after 3 and 10 days of treatment compared with CTRL (+) (*p* < 0.05). However, there were no significant differences between the groups that received SD CUR–BA MP and AA/SD CUR–BA MP (*p* > 0.05), and in the case of turkey poults treated with AA, there were no significant differences with CTRL (+). These results are consistent with other studies published by our research group in which the antimicrobial effect of CUR–BA on *S. enteritidis* counts in cecal tonsils was demonstrated, as well as its synergistic effect due to the formation of complexes that favor the improvement in solubility and stability of curcumin, which allows for its biological properties to be enhanced [13]. 

Furthermore, the results of *S. enteritidis* counts in the crop and cecal tonsils are supported by the body weight gain results from 0 to 10 d, since turkey poults treated with AA/SD CUR–BA MP were the ones that presented the greatest BWG (Table 2), which is related to a lower severity of *S. enteritidis* infection, since *Salmonella* infection is known to be associated with poor nutrient absorption and overall performance in poultry [47]. Likewise, *S. enteritidis* infection can cause intestinal dysbiosis, mild intestinal inflammation, and intestinal barrier damage, along with invasion of internal organs and carcasses of poultry [48]. Therefore, an indirect measure that was used to evaluate damage at the intestinal level was through the serum determination of FITC-d, a molecule that is not permeable under healthy intestinal conditions [49]. The results show that in all treatments, the serum concentration of FITC-d was significantly lower compared with CTRL (+) and in the case of the groups treated with SD CUR–BA MP and AA/SD CUR–BA MP, there were no significant differences compared with CTRL (−), suggesting a decrease in intestinal damage caused by *S. enteritidis*.

## 4. Materials and Methods

### 4.1. Compatibility Studies for Formulation Selection

#### 4.1.1. Preparation of Mixtures

Four mixtures were prepared: (1) ascorbic acid–curcumin (AA–CUR, 1:1 weight ratio), (2) boric acid–curcumin (BA–CUR, 1:1 weight ratio), (3) ascorbic acid–boric acid (AA–BA, 1:1 weight ratio), and (4) ascorbic acid–boric acid–curcumin (AA–BA–CUR, 1:1:1 weight ratio). Briefly, the components of each mixture were homogenized for 10 min. Subsequently, a solution of water–ethanol (1:1 volume ratio) was sprayed to the mixtures in a ratio of 15% with respect to the total weight of the mixture, and the mixing was continued for another 5 min to promote the interactions between the components. Finally, the mixtures were dried for 12 h at 40 °C.

#### 4.1.2. Attenuated Total Reflectance–Fourier Transform Infrared Spectroscopy of the Middle Infrared Region (ATR-FTIR-MIR)

The FT-MIR spectra of each mixture were obtained by means of a Frontier ATR-FT/MIR spectrometer (Perkin Elmer, Norwalk, CT, USA) equipped with a Miracle diamond ATR unit, in a scan range of 450–4000 cm^−1^, with an average of 32 scans and a resolution of 4 cm^−1^. A background spectrum was collected between each sample, and the baseline correction and ATR correction were applied to each spectrum.

#### 4.1.3. Characterization of the Mixtures via Differential Scanning Calorimetry (DSC)

The thermal analysis of each mixture was performed on a “TA Q2000 Modulated DSC” system (TA Instruments, New Castle, DE, USA). Briefly, a sample of each mixture was placed in aluminum pans (3–8 mg) and subsequently hermetically sealed. An aluminum empty pan was used as a reference. DSC analysis started with keeping the samples at 20 °C for 20 min to reach a steady state under a nitrogen atmosphere (flow: 50 mL/min). Then, the samples were equilibrated at 25 °C and heated until reaching a temperature of 220 °C, considering a heating rate of 10 °C/min. The acquisition of the thermograms and the temperature changes/control were programmed in the “TA Instrument Control” software for Q series (version 2.8.394). 

### 4.2. Preparation of the Solid Dispersion of Curcumin–Boric Acid (SD CUR–BA)

SD CUR–BA was prepared in two stages. The first stage consisted of preparing the solid dispersion of curcumin (SD CUR, ≈50%, Mixim Laboratories, Naucalpan, Mexico) by solubilizing 9 parts of polyvinylpyrrolidone K30 (PVP K30, Agrimer^TM^ K-30, Ashland, Columbus, OH, USA) in water and combining 1 part of CUR for its subsequent drying at 40 °C for 48 h. In the second stage, 1 part of SD CUR, 1 part of BA (99.9%, food grade, Drogueria Cosmopolitan, Naucalpan, Mexico), and 10% of a water–ethanol (1:1 weight ratio) solution were sprayed with respect to the total solid dispersion (SD CUR–BA). Finally, the SD CUR–BA was dried at 40 °C for 48 h, followed by the reduction in the particle size using a blade mill (Domccy, Shenzhen, China) and sieving in meshes no. 26 and 28 to achieve an average size of 675 µm.

### 4.3. Obtention of Eudragit FS 30D Microparticles of the Curcumin–Boric Acid Solid Dispersion

SD CUR–BA microparticles were coated with Eudragit^®^ FS 30D (Evonik Rohm GmbH, Darmstadt, Germany) using a Fluid Bed apparatus with a Wurster insert (Mini Glatt 5, Binzen, Germany). Briefly, for the coating dispersion of the microparticles, 8.375 g of Eudragit FS 30D (30% (*w*/*w*) aqueous dispersion) and 3.750 g of PlasACRYL T20 (Evonik Rohm GmbH, Darmstadt, Germany) as a plasticizer, which corresponded to 15% of the total coating dispersion, were added to 12.875 g of water for a total of 25 g of coating dispersion. In the case of the process parameters for obtaining the coated microparticles, 15 g of the SD CUR–BA microparticles (particle size approx. 675 μm) were placed in the fluidization chamber. Subsequently, the parameters were set at an inlet air temperature of 50 °C, fluidization pressure at 0.3 bar, atomization pressure at 0.7 bar, and coating dispersion flow at 0.8–0.9 g/min. Before starting the coating process, the SD CUR–BA microparticles were conditioned at 50 °C for 5 min. The coating process was terminated when a weight gain of 20 g coating dispersion (approx. 17%) was reached. The coated microparticles were kept for a further 20 min at 50 °C to ensure complete drying.

### 4.4. Eudragit FS 30D Microparticle Release Studies

Release studies of Eudragit FS 30D microparticles containing SD CUR–BA (SD CUR–BA MP) were performed in a flow-through dissolution apparatus (USP IV Apparatus, Sotax CH-4008, Basel, Switzerland) equipped with 22.6 mm diameter cells (Figure 5). A 5 mm diameter ruby bead was placed in each of the 22.6 mm cell bases, followed by 6.5 g of a 1 mm diameter glass beads and a 0.7 μm Whatman^®^ glass microfiber filter (GF/F, Millipore-Sigma, Burlington, MA, USA). The dissolution media consisted of 150 mL of degassed buffer solutions of hydrochloric acid pH 1.2, acetates pH 5.2, and phosphates pH 6.8 and 7.0, each at a flow rate of 20 mL/min and 42 °C. The dissolution apparatus was used in a closed-loop configuration considering 12 independent samples of 351 mg Eudragit FS 30D microparticle equivalent to 300 mg SD CUR–BA. The samples were collected manually (2 mL) every 5 min for 20 min, then one more was collected at 30 min in an acetate buffer pH 5.2 (first medium). Subsequently, a pH change was made to 1.2 (second medium) and three samples were collected every 5 min and 15 min, respectively, to reach 45 min of dissolution. Once the dissolution process in the second dissolution medium was completed, a third pH change was made to 6.8 (third medium) and two samples were collected at 5 min, two samples at 10 min and one sample at 15 min to complete a further 45 min of dissolution. Finally, the pH was changed to 7.0 (fourth medium) and a sample was collected at 15 min, and two more samples at 30 min to complete another 75 min. The total release study time was 195 min. In each sample collection, the medium was replaced to maintain sink conditions. The samples were analyzed spectrophotometrically at 421 nm (Varian Cary 1E UV-Vis spectrophotometer, Santa Clara, CA, USA).

### 4.5. Degradation Studies of SD CUR–BA at Different pH

The degradation studies of SD CUR–BA were performed in the USP IV apparatus under the same conditions as for the release studies of SD CUR–BA MP to determine the degradation of SD CUR–BA with respect to time and pH (Figure 5). Briefly, three different proportions of PVP K30 were tested in solid dispersions of CUR, which corresponded to PVP K30 concentrations of 0.3 mg/mL, 0.6 mg/mL, and 1.2 mg/mL. The media used to evaluate the degradation of SD CUR–BA consisted of 150 mL of buffer solutions of hydrochloric acid pH 1.2, acetates pH 5.2, or phosphates pH 6.8 and 7.0 at 42 °C. At 5, 10, 20, 30, 45, 60, 90, and 120 min, the samples were taken with medium replacement. The samples were analyzed spectrophotometrically at 421 nm (Varian Cary 1E UV-Vis spectrophotometer, Santa Clara, CA, USA). Sink conditions were maintained at all times in accordance with previously performed studies [13]. 

### 4.6. In Vivo Experiments

#### 4.6.1. Experimental Groups and Diets

The experimental groups were as follows: (1) Negative control with non-supplemented diet (NC); (2) Positive control with a non-supplemented diet + *S. enteritidis* challenge (PC); (3) Ascorbic acid (0.033%) in the basal diet + *S. enteritidis* challenge (AA); (4) Eudragit FS 30D microparticles (0.067%) of SD CUR–BA in the basal diet + *S. enteritidis* challenge (SD CUR–BA MP); and (5) AA/SD CUR–BA MP (0.1%) in the basal diet + *S. enteritidis* challenge. The basal diet consisted of a starter feed formulation that approximated the nutritional requirements of broilers as recommended by the National Research Council [50] and adjusted to the breeder’s recommendations [51]. No antibiotics, coccidiostats or enzymes were added to the feed (Table 3). All animal handling procedures complied with the Institutional Animal Care and Use Committee (IACUC) at the University of Arkansas, Fayetteville (protocol #15006).

#### 4.6.2. Salmonella Strain and Culture Conditions

A primary isolate from poultry of the bacteriophage type 13A of *Salmonella enterica* serovar *enteritidis* (*S. enteritidis*) was obtained from the USDA National Veterinary Services Laboratory (Ames, IA, USA). This strain is resistant to 25 µg/mL of novobiocin (NO, catalog no. N-1628, Sigma, St. Louis, MO, USA) and 20 µg/mL of nalidixic acid (NA, catalog no. N-4382, Sigma, St. Louis, MO, USA). For the study, 100 µL of *S. enteritidis* from a frozen aliquot were added to 10 mL of tryptic soy broth (TSB, catalog no. 22092, Sigma, St. Louis, MO, USA) and incubated at 37 °C for 8 h, and passed three times every 8 h to ensure that all bacteria were in the log phase [52]. Post-incubation, bacterial cells were washed three times with sterile 0.9% saline by centrifugation at 1864× *g* for 10 min, reconstituted in saline, quantified by densitometry with a spectrophotometer (Spectronic 20DC, Spectronic Instruments Thermo Scientific, Rochester, NY, USA) and finally diluted to an approximate concentration of 4 × 10^4^ cfu/mL. Concentrations of *S. enteritidis* were further verified by serial dilutions and plated on brilliant green agar (BGA, Catalog No. 70134, Sigma, St. Louis, MO, USA) with NO and NA for enumeration of actual colony forming units (cfu) used in the experiment.

#### 4.6.3. Experimental Design

The purpose of the experiment was to evaluate the antimicrobial effect of the treatments in turkeys infected with *S. enteritidis* (therapeutic model). Briefly, 150 one-day-old male turkey poults (Fayetteville, AR, USA) were challenged with 1 × 10^4^ *S. enteritidis* cfu per bird and randomly allocated to one of five groups (n = 30 turkey poults), as described above. Turkey poults were housed in isolation cabins, provided with their respective diet and water ad libitum, and maintained at an age-appropriate temperature during the 10 days of the experiment. At days 3 and 10 post *S. enteritidis* challenge, 15 turkey poults from each group were euthanized by CO_2_ inhalation, and the crop and cecal tonsils (CT) of 12 turkey poults per group were collected aseptically to evaluate *S*. *enteritidis* counts. In addition, blood samples were collected from the femoral vein to obtain serum via centrifugation (1000× *g* for 15 min) to determine the concentration of fluorescein isothiocyanate-dextran (FITC-d) only on day 10 after *S. enteritidis* challenge. The concentration of FITC-d administered was calculated based on the body weight of the experimental group on day 9 after *S. enteritidis* challenge. Likewise, intestinal samples were also collected for total intestinal IgA levels.

#### 4.6.4. Salmonella Recovery

Crop and CT samples from each experimental group on days 3 and 10 after *S. enteritidis* challenge were homogenized (1:4 *w*/*v*) and 10-fold dilutions were plated on Xylose Lysine Tergitol-4 (XLT-4, Catalog No. 223410, BD Difco^TM^) selective media. Bacterial cultures were incubated at 37 °C for 24 h to enumerate total *S*. *enteritidis* cfu. 

#### 4.6.5. Determination of FITC-d in Serum Samples

Fluorescein isothiocyanate-dextran (FITC-d, MW 3–5 kDa; Sigma-Aldrich Co., St. Louis, MO, USA), a marker of enteric permeability and intestinal integrity [53,54], was administrated to 12 turkey poults on day 10 post *S*. *enteritidis* challenge one hour before they were euthanized by CO_2_ inhalation. FITC-d was administered via oral gavage at a dose of 8.32 mg/kg body weight, and three turkey poults per group were used as controls. The serum samples obtained were diluted and measured fluorometrically at an excitation wavelength of 485 nm and an emission wavelength of 528 nm (Synergy HT, Multi-mode microplate reader, BioTek Instruments, Inc., Winooski, VT, USA) to determine the concentration of the FITC-d [55].

### 4.7. Data and Statistical Analysis

Data from *S*. *enteritidis* counts (Log cfu/g), body weight (BW), body weight gain (BWG), serum FITC-d concentration, and total IgA level were subjected to an analysis of variance (ANOVA) as a completely randomized design, using the general linear models procedure of the Statistical Analysis System (SAS) [56]. Significant differences among the means were determined using Tukey’s multiple range test at *p* < 0.05. 

## 5. Conclusions

The selection of appropriate components with antimicrobial activity for the development of new treatments must stop being arbitrary and must be focused on the use of calorimetric and spectrometric techniques, as well as in vitro studies to support the use of each of the components within the treatment (interaction studies). In the present study, the use of a combination of AA and CUR–BA as an integral approach to treat *S. enteritidis* infections showed that it could be a viable option to replace antibiotics, since it can act at the crop and intestine levels. Other studies with higher doses of treatments are being carried out to completely reduce *S. enteritidis* counts in the crop and intestine, using turkey poults as a model. The results will be published elsewhere.

## Figures and Tables

**Figure 1 ijms-24-16186-f001:**
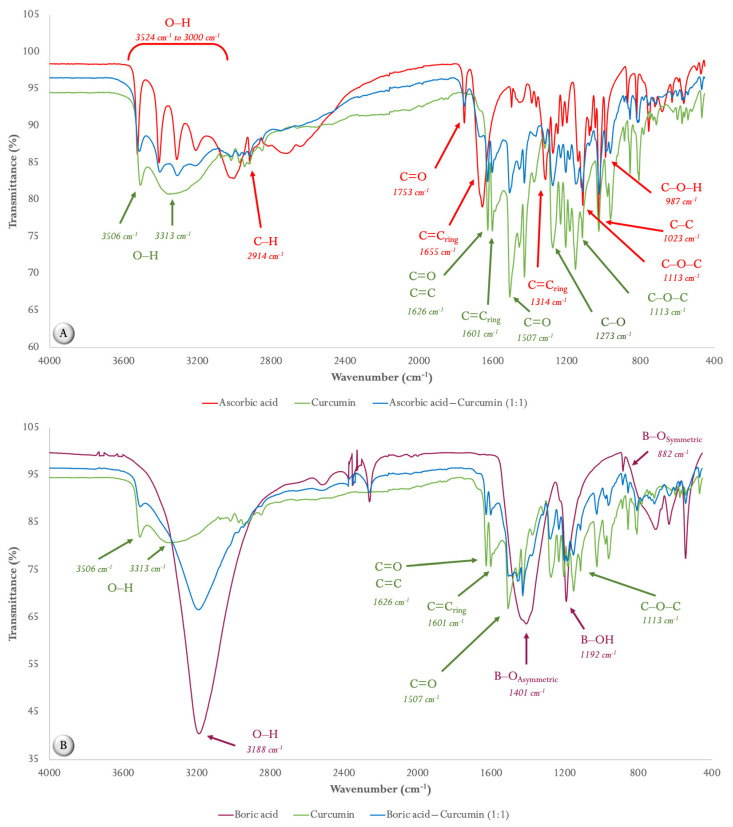
Fourier transform infrared (FTIR) spectra of (**A**) ascorbic acid (AA), curcumin (CUR), and AA–CUR (1:1 weight ratio); (**B**) boric acid (BA), CUR, and BA–CUR (1:1 weight ratio); (**C**) AA, BA, and AA–BA (1:1 weight ratio); and (**D**) AA, BA, CUR, and AA–BA–CUR (1:1:1 weight ratio).

**Figure 2 ijms-24-16186-f002:**
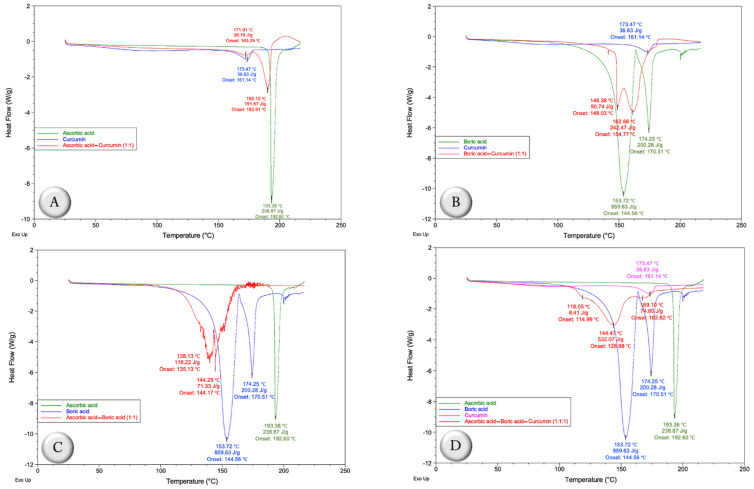
Differential scanning calorimetry (DSC) thermograms of (**A**) ascorbic acid (AA), curcumin (CUR), and AA–CUR (1:1 weight ratio); (**B**) boric acid (BA), CUR, and BA–CUR (1:1 weight ratio); (**C**) AA, BA, and AA–BA (1:1 weight ratio); and (**D**) AA, BA, CUR, and AA–BA–CUR (1:1:1 weight ratio).

**Figure 3 ijms-24-16186-f003:**
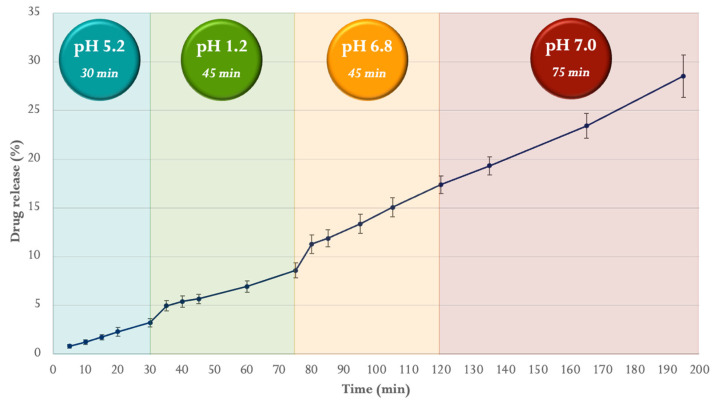
Release profile of SD CUR–BA from Eudragit FS 30D microparticles (SD CUR–BA MP) at pH 5.2, 1.2, 6.8, and 7.2 at 42 °C in the flow-through dissolution apparatus (USP IV Apparatus). Data shown are the mean ± SD, n = 3.

**Figure 4 ijms-24-16186-f004:**
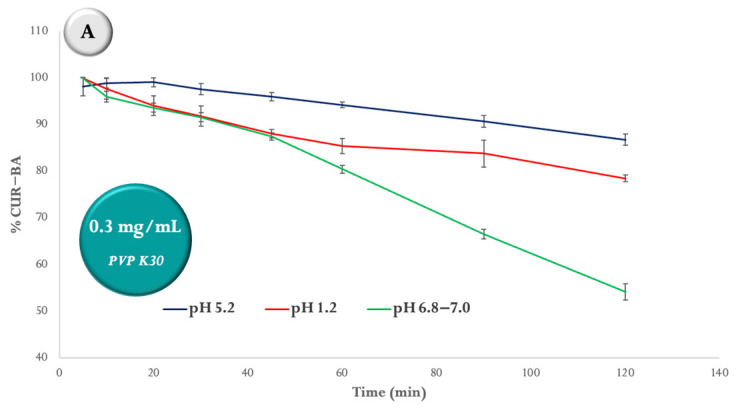
Degradation profiles of SD CUR–BA at pH 5.2, 1.2 and 6.8–7.0 for 120 min at 42 °C in the USP IV apparatus, considering the following concentrations of PVP K30 in the solid dispersion: (**A**) 0.3 mg/mL, (**B**) 0.6 mg/mL, and (**C**) 1.2 mg/mL.

**Figure 5 ijms-24-16186-f005:**
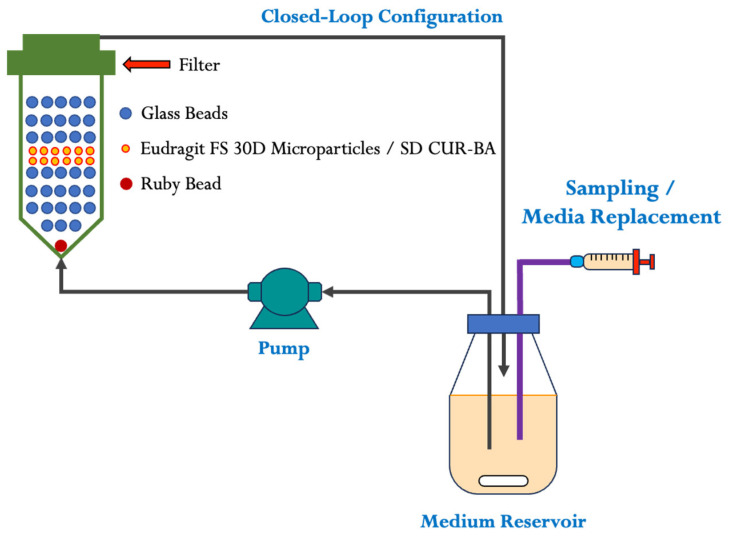
Schematic diagram of the closed-loop configuration for the flow-through cell apparatus (USP IV Apparatus) used for the release studies of Eudragit FS 30D microparticles containing SD CUR–BA (SD CUR–BA MP) and evaluate the stability studies of SD CUR–BA.

**Table 1 ijms-24-16186-t001:** *S. enteritidis* counts ^1^ in crop and cecal tonsils (CT) in turkey poults supplemented with ascorbic acid (AA), Eudragit FS 30D microparticles containing SD CUR–BA (SD CUR-BA MP), and AA/SD CUR–BA MP at three and ten days post-*S. enteritidis* challenge ^2^.

Treatment	Crop Log_10_ cfu/g	CT Log_10_ cfu/g
	3 d of treatment	
CTRL (−)	0.00 ± 0.00 ^b^	0.00 ± 0.00 ^c^
CTRL (+)	3.23 ± 0.14 ^a^	5.76 ± 0.27 ^a^
AA	2.72 ± 0.20 ^a^	5.55 ± 0.19 ^ab^
SD CUR–BA MP	2.96 ± 0.25 ^a^	5.21 ± 0.18 ^ab^
AA/SD CUR–BA MP	3.09 ± 0.16 ^a^	5.16 ± 0.18 ^b^
	10 d of treatment	
CTRL (−)	0.00 ± 0.00 ^c^	0.00 ± 0.00 ^c^
CTRL (+)	3.04 ± 0.27 ^a^	6.00 ± 0.20 ^a^
AA	2.31 ± 0.21 ^b^	5.92 ± 0.38 ^a^
SD CUR–BA MP	2.80 ± 0.42 ^ab^	5.13 ± 0.34 ^b^
AA/SD CUR–BA MP	2.37 ± 0.17 ^ab^	4.97 ± 0.27 ^b^

^1^ Data expressed as log_10_ cfu/g of the tissue. Mean ± standard error from 12 turkey poults. ^2^ Turkey poults were orally gavaged with 10^4^ cfu of *S. enteritidis*. ^a–c^ Different superscripts within treatments indicate significant differences (*p* < 0.05).

**Table 2 ijms-24-16186-t002:** Evaluation of supplementation with ascorbic acid (AA), Eudragit FS 30D microparticles containing SD CUR–BA (SD CUR–BA MP), and AA/SD CUR–BA MP on body weight (BW), body weight gained (BWG), and serum fluorescein isothiocyanate–dextran (FITC-d) levels in turkey poults challenged with *S. enteritidis*
^1^.

Treatment	BW D_0_ (g)	BW D_10_ (g)	BWG (g)	FITC-d (ng/mL)
CTRL (−)	61.13 ± 0.71 ^ab^	184.00 ± 5.00 ^ab^	122.17 ± 5.50 ^ab^	12.76 ± 7.01 ^c^
CTRL (+)	62.07 ± 0.73 ^a^	171.41 ± 4.23 ^b^	108.65 ± 4.60 ^b^	163.38 ± 27.01 ^a^
AA	58.30 ± 0.82 ^bc^	181.89 ± 5.41 ^ab^	123.89 ± 5.70 ^ab^	86.33 ± 12.27 ^b^
SD CUR–BA MP	60.69 ± 0.82 ^ab^	193.87 ± 5.48 ^a^	133.40 ± 5.73 ^a^	58.72 ± 9.58 ^bc^
AA/SD CUR–BA MP	57.33 ± 0.74 ^c^	195.33 ± 6.14 ^a^	137.22 ± 6.41 ^a^	71.48 ± 12.12 ^bc^

^1^ Data are presented as the mean ± standard error from 12 turkey poults. ^a–c^ Different superscripts within treatments indicate significant differences (*p* < 0.05).

**Table 3 ijms-24-16186-t003:** Ingredient composition and nutrient content of the basal starter diet used in the experiments on an as-is basis.

Ingredient	g/kg
Corn	574.5
Soybean meal	346.6
Poultry fat ^1^	34.5
Dicalcium phosphate	18.6
Calcium carbonate	9.9
Salt	3.8
DL-Methionine	3.3
L-Lysine HCL	3.1
Threonine	1.2
Choline chloride 60%	2.0
Vitamin premix ^2^	1.0
Mineral premix ^3^	1.0
Antioxidant ^4^	0.5
Calculated analysis	
Metabolizable energy (MJ/kg)	12.7
Crude protein (g/kg)	221.5

^1^ Poultry fat West Coast Reduction LTD was primarily obtained from the tissue of poultry in the commercial process of rendering or extracting. This finished product was used as an energy source for animal and aquaculture feed. ^2^ Vitamin premix supplied per kg of diet: retinol, 6 mg; cholecalciferol, 150 µg; dl-α-tocopherol, 67.5 mg; menadione, 9 mg; thiamine, 3 mg; riboflavin, 12 mg; pantothenic acid, 18 mg; niacin, 60 mg; pyridoxine, 5 mg; folic acid, 2 mg; biotin, 0.3 mg; cyanocobalamin, 0.4 mg. ^3^ Mineral premix supplied per kg of diet: Mn, 120 mg; Zn, 100 mg; Fe, 120 mg; copper, 10 mg to 15 mg; iodine, 0.7 mg; selenium, 0.2 mg; and cobalt, 0.2 mg. ^4^ Ethoxyquin.

## Data Availability

The data that support the findings of this study are available within this article or from the corresponding author upon reasonable request.

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
