# Peer review of "Evaluation of the Antimicrobial Activity of a Formulation Containing Ascorbic Acid and Eudragit FS 30D Microparticles for the Controlled Release of a Curcumin–Boric Acid Solid Dispersion in Turkey Poults Infected with Salmonella enteritidis: A Therapeutic Model"

_ijms, 2023, doi:10.3390/ijms242216186_

Round 1
Reviewer 1 Report
Comments and Suggestions for Authors
The manuscript is very interesting and important as not always two components act synergistically. The presentation is at a high level; however, some parts need improvement, comments, and corrections. All suggestions are shown in a PDF file (marked in text with comments added).

.
Author Response
We thank you very much for the time you have spent on reviewing our manuscript. We have given full consideration to your comments and the manuscript that has been carefully revised and modified accordingly. Please refer to the point-by-point reply to your comments.
Please see the attachment

Reviewer 2 Report
Comments and Suggestions for Authors
Dear editor,
the paper treats an interesting argument and is in general well written, with some important exception. however some details require more explanations.
l.170: it is not clear to me why the release profiles experiment was engineered as presented, the choice of different time lengths and the origin of those awkward steps in the subplots, which are otherwise remarkably linear in nature.
l.268: "Considering these results, a strong interaction between the components is suggested, which indicates antagonistic interactions between AA and CUR", by no means can IR and DSC give indications about antagonistic interactions between two compounds, whatever the authors mean by antagonistic.
l.357:"The results clearly show that AA has a significant antimicrobial effect at the crop level compared to CTRL (+) at 3 and 10 d of treatment (p < 0.05) and although there were no significant differences in the counts of S. Enteritidis in crop in turkey poults supplemented with AA/SD CUR-BA MP on day 10 of treatment compared to CTRL (+), S. Enteritidis counts in poults supplemented with AA or AA/SD CUR-BA MP did not show significant (p0.05)", so what? there are some of these paragraphs of which I do not understand the deduction.
l.216: "even though the turkey poults that were treated with AA/SD CUR-BA MP had a significantly lower initial weight (p < 0.05) compared to the turkey poults of the other groups ...", this is really odd, how can 150 chickens be divided into 5 groups of 30 random chickens and have all 5 groups differing significantly among themselves and from the mean? The simplest answer is that the reported sigma values are completely wrong.
l.396: "Subsequently, a solution of water - ethanol (1:1) was sprayed to the mixtures in a ratio of 15% with respect to the total mixture and the mixing was continued for another 5 min to promote the interactions between the components.", so, what is the reason of this procedure? why do the authors want to promote interactions when they desire to obtain the separate effects of their compounds. Instead, I think the authors should explain what would be the advantages of an integrated solution over feeding three different products, so minimizing their chemical interactions.
Comments on the Quality of English Language
y-> and
113 -> 1113
(1:1:1) in which units?
Author Response

(The authors gave the same response as above.)

Reviewer 3 Report
Comments and Suggestions for Authors
The current study presented an alternative method to treat and avoid S. Enteritidis infection and future outbreaks in Turkey poultry utilizing successfully a mixture of Eudragit FS 30D microparticles and CUR-BA.
The paper is well-organized and easy to follow.
One thing that I would like to mention:
-According to the description in material and methods, the challenge with S. Enteritidis was performed in Turkeys 150 days old. However, they are mentioned as chickens in the text. Similarly, in the tables' description, and discussion are comparisons mentioned within chickens not as turkeys. This is a bit confusing, whereas there are no chickens described in the in vivo experiments. Please clarify.
-Additionally, in vivo and in vitro are Latin words and should be in italics please revise within the manuscript.
Author Response

(The authors gave the same response as above.)

Round 2
Reviewer 1 Report
Comments and Suggestions for Authors
The manuscript was substantially revised, and the Authors referred to all my comments. There are still some minor mistakes -see attached file. I suggest accepting after the small corrections.

-
Author Response
Dear reviewer,
We greatly appreciate the time dedicated and the suggestions made to improve the quality of this manuscript.
The attached manuscript is the most recent version and addresses the points of the second revision.
Kind regards.

Reviewer 2 Report
Comments and Suggestions for Authors
Dear editor,
I am currently satisfied with the state of the paper. I am still not convinced that the randomization process on the samples was conducted correctly: let's just say that winning the lottery would be easier than obtaining those five groups of thirty chickens.
l.346 and l.349: the same phrase is repeated twice.
best wishes
Author Response
Dear reviewer,
We greatly appreciate the time dedicated and the suggestions made to improve the quality of this manuscript.
The attached manuscript is the most recent version and addresses the points of the second revision.
Kind regards
Daniel
